# Improved Enamel Acid Resistance Using Biocompatible Nano-Hydroxyapatite Coating Method

**DOI:** 10.3390/ma15207171

**Published:** 2022-10-14

**Authors:** Ryouichi Satou, Miyu Iwasaki, Hideyuki Kamijo, Naoki Sugihara

**Affiliations:** 1Department of Epidemiology and Public Health, Tokyo Dental College, Tokyo 101-0061, Japan; 2Department of Social Security for Dentistry, Tokyo Dental College, Tokyo 101-0061, Japan

**Keywords:** apatite, fluoride, enamel, demineralization, preventive dentistry

## Abstract

In this study, we attempted to develop a dental caries prevention method using a bioapatite (BioHap), an eggshell-derived apatite with nanoparticle size and biocompatibility, with a high-concentration fluoride tooth surface application method. The enamel acid resistance after the application of the proposed method was compared with that of a conventional topical application of fluoride using bovine tooth enamel as an example. The tooth samples were divided into three groups based on the preventive treatment applied, and an acid challenge was performed. The samples were evaluated for acid resistance using qualitative and quantitative analytical methods. The BioHap group demonstrated reduced enamel loss and improved micro-Vickers hardness, along with a thick coating layer, decreased reaction area depth, and decreased mineral loss value and lesion depth. The combination of BioHap with high-concentration fluoride led to the formation of a thick coating layer on the enamel surface and better suppression of demineralization than the conventional method, both qualitatively and quantitatively. The proposed biocompatible nano-hydroxyapatite coating method is expected to become a new standard for providing professional care to prevent dental caries.

## 1. Introduction

Apatite is a bioceramic used in the medical and dental fields for various treatments, such as the construction of artificial bones and surface coating of implants [1,2,3]. Hydroxyapatite (HAp), represented as Ca_10_(PO_4_)_6_(OH)_2_, is the main component of vertebrate bones and teeth; it constitutes approximately 70, 97, and 70% (weight %) of bone, tooth enamel, and dentin, respectively [4,5]. Synthetic apatites, such as HAp and amorphous calcium phosphate, have become popular as materials for repairing enamel damage because their chemical compositions are similar to that of tooth enamel [1,2]. Recently, HAp has been used in biomaterials, as well as in preventive dentistry applications, such as toothpastes and mouthwashes [6,7,8,9].

Dental caries, a disease caused by the dissolution of enamel and dentin HAp by the acid produced by bacteria on the tooth surface [10], can be prevented by employing a tooth surface application and mouth rinse methods in dental clinics; moreover, the use of fluoride-containing dentifrices at home can effectively strengthen the tooth structure [11,12,13]. A fluoride tooth surface application method is commonly used for preventing dental caries; however, some reports have suggested the combined use of high-concentration fluoride and HAp in professional care. Dentifrice, a low-concentration fluoride application method used in home care, which has been reported to promote remineralization and strengthen tooth structure owing to its combined use of fluoride and HAp, is being commercialized [9,14,15]. A systematic review of 219 randomized controlled trials evaluating the risk of caries in oral care products showed that the inclusion of HAp reduced caries by 17% [6].

Therefore, we attempted to develop a new caries prevention method by combining HAp, which can effectively prevent caries, with a high-concentration fluoride tooth surface application method in professional care. According to the Pharmaceutical Affairs Law of each country, the concentration of fluoride in toothpaste is as low as 500–1500 ppm. However, a high concentration of 9000 ppm or more can be used in fluoride tooth surface application methods performed by dentists [10,16,17,18]. Many clinical studies have demonstrated that, depending on the dose, fluoride increases the caries prevention effect [16,17,18]. Fluoride toothpaste and mouth rinse require multiple daily applications; however, professional fluoride treatments only need to be administered once every six months or each year to ensure protection against caries [10].

In the proposed method, we select bioapatite (BioHap, BIOAPATITE Inc., Shiga, Japan) based on its biocompatibility and particle size [19]. BioHap is characterized by a significantly small particle size of 20–50 nm; its composition is also rich in magnesium ((Ca:Mg)_10_(PO_4_)_6_(OH)_2_; Japanese Laid-Open Patent Publication No. 2020-105060 and 2020-117423). Schroeder et al. reported that Mg^2+^ promoted the formation of whitlockite, which is required for HAp conversion [20,21]. Moreover, it is reported to be approximately four times more biocompatible than mineral-derived apatite, although it is still not widely used in dental practice [22,23]. In this study, we attempt to develop a new dental caries prevention method combining BioHap with a high-concentration fluoride tooth surface application. We also compare enamel acid resistance after the application of the proposed method with that of a conventional topical application of fluoride.

## 2. Materials and Methods

### 2.1. Preparation of Enamel Samples

A total of 24 bovine anterior mandibular teeth were used in this study. After removing the attached gingiva and cement, the root of each tooth was removed, and only the crown enamel was used in the experiment. Subsequently, enamel blocks (1 cm × 1 cm × 1 cm) were prepared and mirror-polished with water-resistant abrasive paper (#1000, #2000, and #4000).

### 2.2. Fluoride Application and pH-Cycling Acid Challenge Experiment

The method employed by Miki et al. was used to apply BioHap to the tooth surface enamel [24]. After ionizing some crystals on the surface by applying citric acid gel (5% citric acid, 0.01% malic acid, 5% glycerin, 3% tamarind gum, pH 3.0; AIWA Co., Ltd., Osaka, Japan), BioHap (25 g, BIOAPATITE Inc., Shiga, Japan) and citric gel were mixed in a 1:2 weight ratio and applied to the surface (Table 1). BioHap is synthesized from industrial waste eggshells and its safety has been proved through oral toxicity and human patch tests (Japan Food Research Laboratories: No. 18028586001-0101).

The samples were divided into three groups: (1) untreated fluoride (control, 0 ppmF, pH 7.0), (2) acidulated phosphate fluoride (APF, 9048 ppmF, pH 3.6) for 4 min, and (3) BioHap with APF (BioHap, 9048 ppmF, pH 3.6) for 4 min. Eight samples were prepared from each group (n = 8). To create experimental and control surfaces on the same enamel surface, half of the mirror-polished enamel surface was coated with dental wax (Inlay Wax Soft, 27B2X00008000028, GC Co., Ltd. Tokyo, Japan); further, a pH-cycling test was conducted. After fluoride application for 4 min, all samples from the three groups were immersed in a remineralization solution (0.02 M HEPES-based buffer solution; Ca: 3 mM, P: 1.8 mM, pH 7.3, DS: 10) for 1 h at 37 ℃. Then, the samples were immersed in a demineralization solution (0.1 M lactic acid buffer solution; Ca: 3 mM, P: 1.8 mM, pH 4.5, DS: 10) for 24 h at 37 ℃; each cycle was repeated three times.

### 2.3. Three-Dimensional Laser Microscopic Observation

The samples were dehydrated in an ascending ethanol series after wax removal. We used a three-dimensional (3D) measurement laser microscope (LEXT OLS4000, Olympus Corp., Tokyo, Japan) to measure the step height profile between the experimental surface (ES) and reference surface (RS) after the acid challenge. The amount of substantial tooth defects due to the acid challenge was recorded. The measurement area was 645 × 645 µm; the boundary between the acid-demineralized ES and the wax-protected RS was photographed. Finally, 3D measurements were performed at five sites for each sample, and the mean and standard deviation were determined.

### 2.4. Micro-Vickers Hardness Measurement

The micro-Vickers hardness values were measured using a hardness tester (HMV-1; Shimadzu Corp., Tokyo, Japan) after the samples were dehydrated. The indentation load and time were set to 0.49 N and 20 s, respectively, to measure the Vickers hardness (HV). To compensate for errors due to individual differences in samples, changes in HV before and after the experiment (∆HV = RS − ES) were calculated. The HV and ∆HV values were measured at five locations per sample; then, the mean ± standard deviation values were calculated.

### 2.5. Cross-Section and Surface Morphology through Scanning Electron Microscopy

After the acid challenge, each sample was washed with xylene. After carbon vapor deposition was performed for the analyte sample surface, the tooth surface was observed using a scanning electron microscope (SU6600, HITACHI Ltd., Tokyo, Japan) at an accelerating voltage of 15 kV. The samples were then embedded in polyester resin (Rigolac, Nisshin EM, Tokyo, Japan) to prepare polished sections, and the cross-sections were observed.

### 2.6. Reaction Area Depth Measurement Using Polarizing Microscope

The method employed by Ogata et al. was used to prepare and analyze the enamel sections using polarizing microscopy [7]. The resin-embedded sample used for scanning electron microscopy (SEM) observation was sliced to a thickness of 120 μm and, finally, ground to a thickness of 100 μm using an internal annulus saw microtome (Leica 1600, Leica Microsystems, Heidelberg, Germany). Images were acquired using a polarizing microscope (ECLIPSE E600 Pol, Nikon Corp., Tokyo, Japan) and microscope camera (Axiocam ERc 5s, Carl Zeiss Co., Ltd., Jena, Deutschland); the reaction area depth (RAd, μm) was measured using image analysis software (ZEN lite, Carl Zeiss Co., Ltd., Jena, Deutschland). The RAd was determined as the reaction area from the enamel surface to the deepest line that exhibited discoloration owing to changes in the refractive index caused by demineralization on the images; it was measured at five locations per sample, and the mean ± standard deviation values were calculated.

### 2.7. Contact Microradiography (CMR)

The samples were embedded in polyester resin (Rigolac, Nisshin EM, Tokyo, Japan) to prepare polished sections of 100 µm thickness. A soft X-ray generator (CMR-3, Softex, Tokyo, Japan) equipped with a 20 µm thick Ni filter was used to set the imaging conditions for an aluminum step wedge (20 steps, 20 µm) to enable differentiation from steps 1 to 20. Thus, imaging was performed with a tube voltage, tube current, and radiation time of 15 kV, 3 mA, and 12 min, respectively; further, light microscopy was performed at 200× magnification. A glass plate (High Precision Photo Plate, HRP-SN-2; Konica Minolta, Tokyo, Japan) was used for imaging; it was placed in a developer (D-19: Kodak, Rochester, NY, USA) at 20 °C for 5 min. Then, the plate was fixed for 5 min and washed with water for 10 min. The completed plate was converted to gray scale using image analysis software (Image Pro Plus, version 6.2; Media Cybernetics Inc., Silver Spring, MD, USA) and an image analysis system (HC-2500/OL; OLYMPUS Corp., Tokyo, Japan); consequently, the concentration profile was acquired. The mineral loss value (ΔZ) and lesion depth (Ld) were determined, and the extent of demineralization was compared. All five sites were measured in the range of 50 × 150 µm from the surface vicinity to deep, healthy enamel. For ΔZ, the mineral equivalent was calculated with the formula proposed by Angmar et al. [25] using the sample density and aluminum step wedge (captured simultaneously as reference). The values were converted to a histogram, where the mineral value and healthy enamel sections were 0 and 100%, respectively. Ld was defined as the distance from the enamel surface to a lesion location with mineral content higher than 95% in comparison to the sound enamel [26].

### 2.8. Statistical Analysis

The means ± standard deviations of eight samples were determined to compare the three fluoride applications. The *p*-values were also calculated using one-way analysis of variance (ANOVA); the results were considered significant at *p* < 0.01. The Bonferroni test was used for post hoc comparisons. Graphs were prepared and data were analyzed using Origin software (ORIGIN 2022, Lightstone Corp., Tokyo, Japan).

## 3. Results

### 3.1. Alpha-Tricalcium Phosphate (α-TCP) and BioHap Particle Characterization

Figure 1 shows SEM images of α-TCP and BioHap particles, which have been clinically used in the fields of medicine and dentistry. The particle size of α-TCP was 5–15 μm; it was a massive crystal with multiple interconnected rectangular and polygonal particles (Figure 1a). The crystals of BioHap were smaller than those of α-TCP, and the size of the agglomerated clumps was irregular (Figure 1b). The 5000× magnified image shows that α-TCP particles were composed of multiple polygonal particles with sharply angled crystal edges (Figure 1c). In contrast, the 5000× magnified BioHap image shows amorphously scattered spherical particles with small particle sizes (Figure 1d). In the 15,000 × magnification of α-TCP, the entire crystalline body could not fit into the field of view for observation (Figure 1e). Further, BioHap exhibited amorphous, loosely bound spherical particle aggregates, and the diameter of the constituent particles was significantly smaller than that of α-TCP (Figure 1e,f).

### 3.2. Height Difference Profiles Using 3D Laser Microscope

Figure 2 presents a graph and images of height difference profiles obtained using a 3D laser microscope after the acid challenge. The left side of Figure 2a–c shows the RS protected by wax, which was not demineralized; the right side shows the demineralized ES. In the control group (no fluoride), the ES was significantly demineralized, and a defect of 6.184 ± 0.143 μm was observed on the enamel surface (Figure 2a). In the APF group, the difference in height between the RS and ES decreased to 1.761 ± 0.212 μm, and significant demineralization suppression was confirmed in comparison to the control group (*p* < 0.05, Figure 2b,d). The amount of substantial defects in the BioHap group was 0.859 ± 0.139 μm, which was significantly smaller than that in the APF group (Figure 1c,d). The amount of substantial defects in the BioHap group was the smallest (*p* < 0.05, Figure 1d).

### 3.3. Micro-Vickers Hardness Measurement after Acid Challenge

Figure 3 shows graphs of the micro-HV after the acid challenge. The control group demonstrated an HV of 18.874 ± 6.302 HV, which was significantly smaller than those of the APF and BioHap groups (*p* < 0.05, Figure 3a). The HV values of the APF and BioHap groups improved to 91.817 ± 30.881 and 166.415 ± 57.496 HV, respectively (Figure 3a). A significant difference was observed between the APF and BioHap groups; the BioHap group exhibited the highest value (*p* < 0.05, Figure 3a).

Figure 3B depicts the ΔHV after the acid challenge; the values of ΔHV for the control, APF, and BioHap groups were 329.923 ± 31.482, 231.280 ± 90.559, and 138.496 ± 70.150 HV, respectively (Figure 3b). The control group demonstrated the highest value, which was significantly larger than that of the BioHap group (*p* < 0.05, Figure 3b).

### 3.4. SEM Images of Enamel Surface and Cross-Sections after Acid Challenge

Figure 4 presents secondary electron images of the enamel surface and cross-sections after the acid challenge. The surface image of the control group demonstrated demineralization from the center of the enamel rod, as well as disturbance in the rod arrangement due to acid; a cavitation phenomenon and irregularities between the enamel gap were also observed (Figure 4a). In the APF group, clear enamel rods and gaps were observed (Figure 4b); numerous fine, spherical particles adhered to the surface of the APF group, thereby covering it evenly (Figure 4b). The surface of the BioHap group was covered by a translucent coating layer; consequently, the bumpiness of the enamel gap disappeared, and spherical particles were generated unevenly on the coating layer (Figure 4c). The spherical particles produced in the BioHap group were larger in size and less perfectly spherical in shape than those in the APF group (Figure 4c).

Figure 4d–f presents secondary electron SEM images of cross-sectioned surfaces after the acid challenge. In the control group, the signal intensity and expansion in enamel gaps decreased in the surface-layer range of 0–50 μm. In particular, severe demineralization was observed at 25–50 μm below the surface layer of the control group, accompanied by a collapse in the rod structure (Figure 4d). In the APF group, a thin, acid-resistant layer of 1–2 μm was observed on the surface layer; moreover, a narrow range of enamel rod structure disappeared at 15–20 μm below the surface layer (Figure 4e). The APF group demonstrated less demineralization than the control group (Figure 4e). Moreover, the coating layer of the BioHap group contained numerous spherical particles deposited on the surface (Figure 4f). Demineralization of 2–3 μm and enamel gap expansion were observed just below the coating layer of the BioHap group; however, the signal intensity did not decrease at a depth of 10 μm or more below the surface layer; it is worth noting that the signal intensity was similar to that observed in healthy regions (Figure 4f).

### 3.5. Polarizing Microscope Image of Enamel Cross-Sections after Acid Challenge

Figure 5 shows graphs and a RAd image obtained using a polarizing microscope after the acid challenge. In the control group, yellow and blue polarized regions were uniformly observed from the surface to the deep layer of the enamel, with a RAd of 116.804 ± 21.816 μm (Figure 5a). The RAd of the APF group decreased to 78.834 ± 5.391 μm; moreover, the blue polarized region increased (Figure 5b). The RAd of the APF group was significantly smaller than that of the control group (*p* < 0.01, Figure 5b,d). Moreover, the RAd of the BioHap group was significantly lower than that of the APF group (52.618 ± 3.868 μm; *p* < 0.01) (Figure 5c,d).

### 3.6. Mineral Loss Value and Lesion Depth by CMR Analysis

Figure 6 shows CMR images after the acid challenge, along with a graph that depicts the changes in mineral loss (ΔZ, vol% μm) in each group according to the depth from the enamel surface. In the control group, a thin region with high signal intensity was observed in the surface layer at 0–15 μm; the signal intensity decreased at 15–80 μm owing to structural destruction and demineralization (Figure 6a). In the APF group, a mineral-rich region was observed at 25–50 μm from the surface layer; the appearance of a demineralized layer was confirmed at 50–75 μm below this region (Figure 6b). In the BioHap group, the signal intensity was similar to that observed in healthy enamel, except for demineralization near the surface layer (Figure 6c). The graph depicting the transition of ΔZ depending on depth from the enamel surface layer in each group demonstrates that the rise in the graph was gentle in the control and APF groups and steep in the BioHap group (Figure 6d). ΔZ at 25 μm was low at 25.368 vol% μm in the control group, but it increased to 70.541 vol% μm in the APF group and rapidly increased to 88.688 vol% μm in the BioHap group (Figure 6d).

Figure 7 shows the ΔZ and Ld in each group through CMR analysis. In the control group, ΔZ was 9611.756 ± 1472.511 vol% μm, which was significantly higher than those in other groups (*p* < 0.01, Figure 7a). Moreover, ΔZ decreased to 3419.693 ± 758.731 and 2097.785 ± 339.379 vol% μm in the APF and BioHap groups, which was approximately one-third and one-fifth of that observed in the control group, respectively. However, no significant difference was observed between the BioHap and APF groups (*p* > 0.01, Figure 7a). The control group exhibited the largest Ld, 103.673 ± 5.324 μm, which was significantly higher than that of the BioHap group (*p* < 0.01, Figure 7b). The APF and BioHap groups demonstrated Ld values of 87.528 ± 3.488 and 51.370 ± 10.678 μm, respectively. The BioHap group had the lowest Ld, significantly different from the APF group (*p* < 0.01, Figure 7b).

## 4. Discussion

### 4.1. Enamel Acid Resistance Effect and BioHap Mechanism

The BioHap group formed a 5 μm thick coating of aggregated spherical particles on the enamel surface layer; moreover, it inhibited demineralization better than the APF group. SEM images comparing BioHap particles with α-TCP for clinical dental use showed that the BioHap particles were small in size, less than 50 nm, and amorphous (Figure 1f). It should be noted that small primary particle size of biomaterials is synonymous with larger surface area, which increases the reaction rate during chemical reactions [1,5]. Fulmer et al. reported that, as the particle size decreased, the solubility of apatite and the amount of CaF_2_ produced in the oral cavity increased, thus improving the acid resistance [27,28,29]. α-TCP (α-TCP-A, pulverized product, CAS:7758-87-4; Chemical Industrial Co., Ltd., Osaka, Japan) used in artificial bones and implants has a particle size and theoretical surface area of 8–15 μm and 0.1–2.0 m^2^/g, respectively [1,5]. β-TCP (β-TCP-100, pulverized product, CAS:7758-87-4; Chemical Industrial Co., Ltd., Osaka, Japan), which is smaller than α-TCP, has a particle size and theoretical surface area of 1–5 μm and 1.0–5.0 m^2^/g, respectively [1,5]. Moreover, the particle size of HAp (HAP-100, CAS:1306-06-05; Taihei Chemical Industrial Co., Ltd., Osaka, Japan), which is used as a nanoparticle apatite in dentistry, is less than 1.7 μm, with a theoretical surface area of 50 m^2^/g [1,5,30]. However, the particle size of BioHap observed in this study was 50 nm or less, with a surface area as large as 70 m^2^/g (Japanese Laid-Open Patent Publication No. 2020-105060 and 2020-117423, Figure 1F). Previous studies that have employed combinations of α-TCP and fluoride have reported the formation of very thin coatings of 1–2 μm on the enamel surface layer, but the thick coating layer in this study has not been observed [24,31,32,33]. Thus, the small particle size and large surface area of BioHap might affect the formation speed and thickness of the coating layer.

In this study, 3D laser microscope measurement, a micro-HV test, SEM observation, and polarizing microscope photography were performed for qualitative analysis (Figure 2, Figure 3, Figure 4 and Figure 5). The amount of substantial defects and micro-HV revealed improvement in enamel acid resistance in the BioHap group (Figure 2 and Figure 3). Moreover, the amount of enamel loss in the BioHap group was approximately half of that in the APF group, which suggests that the coating layer formed on the surface had a strong inhibitory effect on demineralization (Figure 2D). The micro-HV of the BioHap group was approximately twice that of the APF group, which indicated higher acid resistance in the BioHap group (Figure 3a). In addition, HV was used for the qualitative evaluation of enamel demineralization; it has also been used as an index of acid resistance in previous studies [34,35]. The HV of healthy bovine tooth enamel is 200–300 HV; demineralization has been reported to change the tooth microstructure and reduce the hardness [34]. The micro-HV of the BioHap group in this study was 166.415 ± 57.496 HV, which was similar to that observed in healthy teeth, even after acid demineralization, in comparison to the control and APF groups (Figure 3a). Cross-sectional SEM images also supported that the coating layer of the BioHap group exhibited acid resistance and maintained the enamel rod structure after the acid challenge (Figure 4f). Regarding the coating layer, Alencar et al. reported that casein phosphopeptide amorphous calcium phosphate and nano-hydroxyapatite (nHAp) applied to the tooth surface formed a coating on the surface layer and inhibited demineralization, similar to this study [31,36]. Soares et al. performed a micro-energy dispersive X-ray fluorescence analysis after applying nHAp and APF gels to the enamel and found that the nHAP-treated group had higher calcium and phosphorus contents than the APF group [31]. The coatings in our study are consistent with those observed in previous studies; moreover, the thickness in the BioHap group was greater than those observed in previous studies (Figure 4f) [24,31,32,33]. The coating layer is expected to physically protect the tooth surface from acid, and it has been reported to suppress tooth demineralization [37,38]. In the surface SEM images obtained in this study, a fragile enamel gap was observed in the APF group, which was blocked by the coating in the BioHap group (Figure 4b,c). This SEM observation suggests that the coating layer acted as a physical barrier against acid. Calcium phosphate has been used as a coating component in previous studies; it undergoes repeated dissolution and reprecipitation to convert into HAp, thus restoring the minerals lost in demineralization [8,39]. In this study, the coating layer recovered the minerals, and demineralization decreased in the BioHap group, as observed in cross-sectional SEM images; moreover, a signal with the same intensity as that in the healthy region was observed at 10 μm below the surface layer (Figure 4f). RAd measurements through polarizing microscopy also showed that demineralization was not observed in deep layers in the BioHap group (Figure 5c,d). The change in color tone in the BioHap group was limited to the surface layer, and no subsurface demineralization image or multi-layered structure due to the migration of minerals was observed, unlike the control and APF groups (Figure 5a–c). These results suggest that the coating of the BioHap group had high acid resistance even after the acid challenge, which physically prevented acid penetration in the enamel. 

In this study, we measured ΔZ and Ld through CMR imaging as a quantitative analysis method (Figure 6 and Figure 7). A large, demineralized image was observed in the control group; ΔZ values in the APF and BioHap groups were reduced to approximately one-third and one-fifth of that in the control group, respectively (Figure 6a and Figure 7a). In addition, the Ld in the BioHap group was approximately half of that in the control group (Figure 7b). The demineralization pattern in the APF group is consistent with those observed in previous studies; however, localized demineralization under the coating layer, such as that in the BioHap group, has not been observed in previous studies [24]. The CMR image of the BioHap group quantitatively clarified that demineralization did not extend to deep layers in the BioHap group; instead, it was observed just below the coating layer and enamel surface.

With respect to the formation mechanism of the coating layer, Miki et al. considered that the calcium ions released from α-TCP mix with citric acid combined with fluoride ions to precipitate CaF_2_ [24]. Huang et al. reported that calcium phosphates, such as dicalcium phosphate dihydrate and octacalcium phosphate, are produced along with CaF_2_ [29,40]. Because the composition of BioHap [(Ca:Mg)_10_ (PO_4_)_6_ (OH)_2_] is similar to that of HAp and α-TCP, it is considered to form a coating layer through a similar mechanism. Fluoride ions are known to form crystal nuclei and grow apatite crystals [16,17,41,42]. We believe that the high-concentration fluoride in this study promoted the formation of calcium fluoride and chemical bonding with enamel crystals. The mixing of citric acid and BioHap in this method generated large amounts of calcium and phosphate ions. This manipulation could rapidly increase the saturation of calcium and phosphorus in the microenvironment near the enamel surface. Mg^2+^ in BioHap exists in the apatite structure replacing Ca^2+^. Therefore, the coating formed with BioHap and fluoride became chemically unstable due to the partial replacement of Ca^2+^ in apatite with Mg^2+^ and demineralized preferentially over teeth during acid stimulation. Until the supersaturated ions in the coating layer and the surrounding enamel surface were completely lost, maintaining the saturation of the microenvironment was considered to contribute to demineralization suppression [18].

### 4.2. Safety of the Proposed Method and Application in Preventive Dentistry

The biological safety of apatite has been proved through cytotoxicity, sensitization, irritation, acute toxicity, and genotoxicity tests [1,5]. The biotoxicity of fluoride ions is a reported issue in the application of fluoride to children [16]. The APF solution used in this method had a high fluoride concentration of 9000 ppmF, but it is the first-choice treatment for dental caries prevention in clinical practice, and safety for children is guaranteed. However, considering the biotoxicity of fluoride ions, it is necessary to ensure that treatment is performed by a dentist, and thorough confirmation of emergency measures and prevention of accidental ingestion is required. In our new method, most of the free fluoride ions were considered to form a coating and combine with calcium ions to form CaF_2_, which is chemically stabile [16,17]. Calcium fluoride has low biotoxicity; moreover, 90 and 60% of calcium fluoride is excreted via the urine within 24 h in adults and children, respectively [10,16]. Therefore, this method was considered to be safer than conventional fluoride tooth surface application methods, and it could be used among children and elderly people with minimal saliva secretion.

Although HAp exhibits excellent biocompatibility and physicochemical properties, it has low mechanical strength (compressive strength: 500–700 MPa; tensile strength: 100–200 MPa; elastic modulus: 50–100 GPa, which is greater than the 4–10 GPa of bone), which makes it a hard and brittle material [1,5,43]. BioHap is similar to HAp and is considered to have similar properties. It should be noted that strength experiments were not performed in this study; thus, we believe that further analysis is required to verify the strength and durability exhibited by the coating layer obtained in this method. The tooth surface is subjected to a wear load owing to eating and brushing 2–3 times a day, along with demineralization due to acid [3]. Although saliva has a strong buffering capacity, the durability of the coating layer of the BioHap group in the oral cavity is unknown. For clinical application, further experiments must be conducted to determine the acid resistance in vivo and coating strength against brushing stimulation. Moreover, recall interval guidelines should be created for patients considering these experiments. This method physically sealed the teeth and could be used not only for dental caries, but also for erosion, such as bioglass paste and resin [44].

## 5. Conclusions

The combination of BioHap and high-concentration fluoride led to the formation of a thick coating layer on the enamel surface, which suppressed demineralization more than the conventional method. The BioHap group exhibited reduced enamel loss and improved micro-HV, along with a thick coating layer and decreased RAd, mineral loss, and Ld. The BioHap group demonstrated higher inhibition of demineralization than the APF group, both qualitatively and quantitatively. These results indicated that BioHap was a beneficial biomaterial in preventive dentistry and that it protected enamel from acid when used in combination with fluoride. The proposed biocompatible nano-hydroxyapatite coating method is expected to become a new standard for providing professional care to prevent dental caries.

## Figures and Tables

**Figure 1 materials-15-07171-f001:**
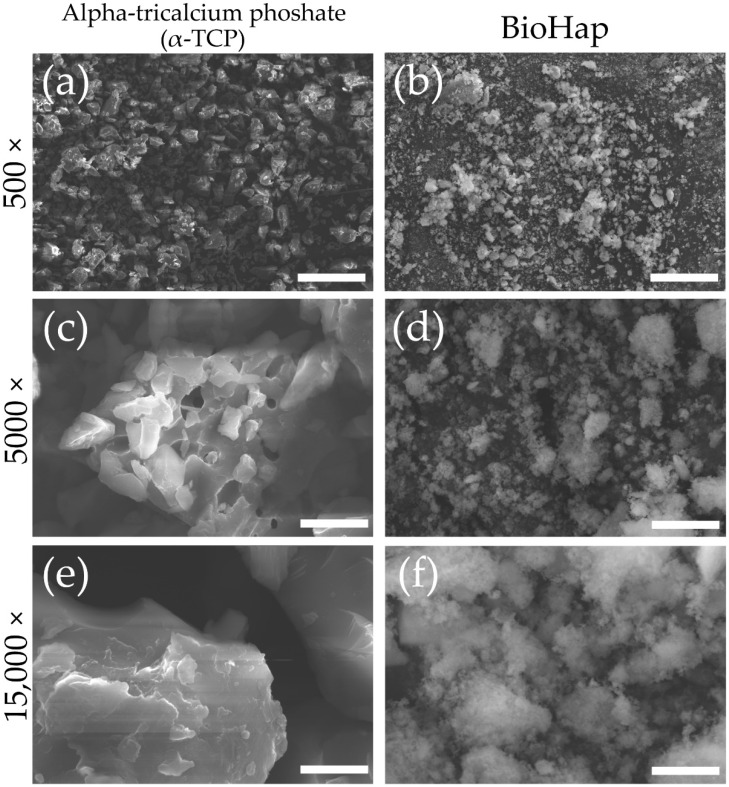
Particle shapes of alpha-tricalcium phoshate (α-TCP) and BioHap. (**a**,**c**,**e**) Scanning electron microscopy (SEM) images of alpha-tricalcium phosphate (α-TCP) particle carbon deposition sample. (**b**,**d**,**f**) SEM images of BioHap particle carbon deposition sample. (**a**,**b**) Scale bar is 50 μm (at 500× magnification). (**c**,**d**) Scale bar is 5 μm (at 5000× magnification). (**e**,**f**) Scale bar is 1.5 μm (at 15,000× magnification).

**Figure 2 materials-15-07171-f002:**
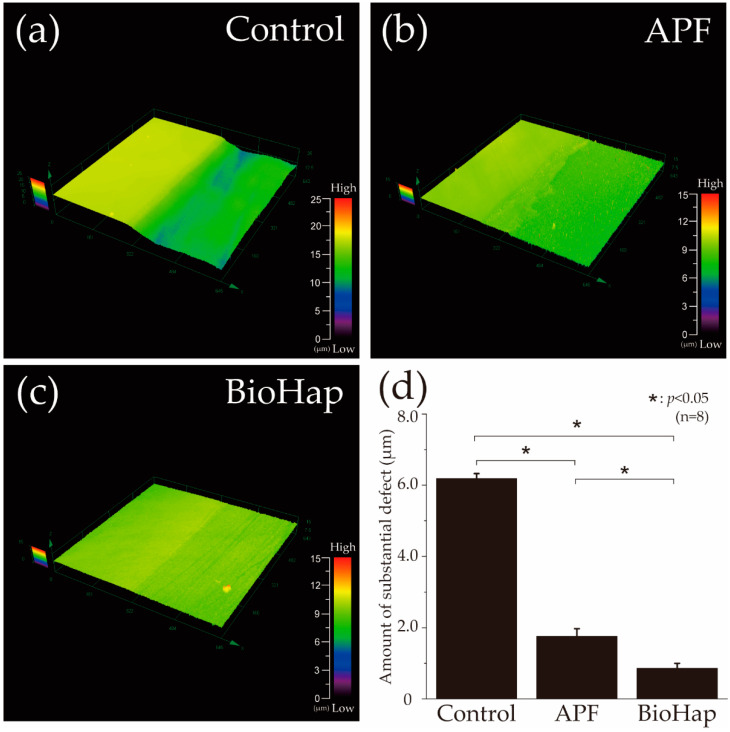
Height difference profiles measured using 3D laser microscope. (**a**) Boundary images of reference and experimental surfaces after acid challenge in the control (not fluoride-treated), (**b**) acidulated phosphate fluoride (APF, 9048 ppmF, pH 3.6), and (**c**) BioHap with APF (9048 ppmF, pH 3.6) groups. The left side in a–c shows the reference surface protected by wax and not demineralized; the right side shows the demineralized experimental surface. (**d**) Graphical representation of substantial defects due to demineralization (n = 8, *: *p* < 0.05).

**Figure 3 materials-15-07171-f003:**
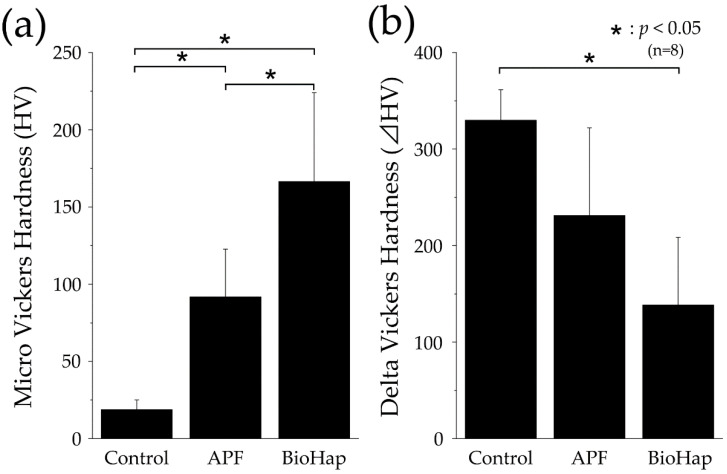
Micro-Vickers hardness (HV) measurements. (**a**) Graph of micro-HV values after acid challenge (n = 8, * *p* < 0.05). (**b**) Graph of ΔHV values (differences in HV values between the reference and experimental surfaces; n = 8, * *p* < 0.05).

**Figure 4 materials-15-07171-f004:**
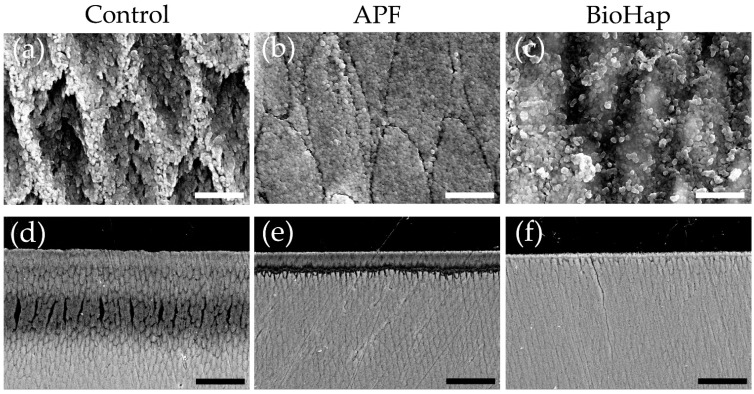
SEM images of enamel surface and cross-sections after acid challenge. Surface SEM images of the (**a**) control, (**b**) APF, and (**c**) BioHap groups. Scale bar is 2.5 μm. All images were recorded at 10,000× magnification carbon deposition samples. Cross-sectional SEM images of the (**d**) control, (**e**) APF, and (**f**) BioHap groups. Scale bar is 25 μm. All images were recorded at 1000× magnification carbon deposition samples.

**Figure 5 materials-15-07171-f005:**
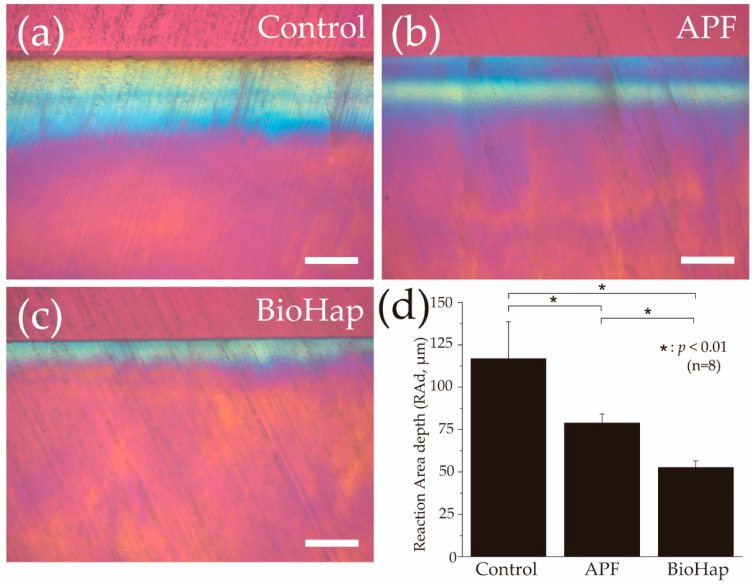
Polarizing microscope images of enamel cross-sections after acid challenge. Polarizing microscope cross-sectional images of the experiment surface after acid challenge in the (**a**) control, (**b**) APF, and (**c**) BioHap groups. Scale bar is 100 μm. (**d**) Graphical representation of reaction area depth due to demineralization (n = 8, * *p* < 0.01).

**Figure 6 materials-15-07171-f006:**
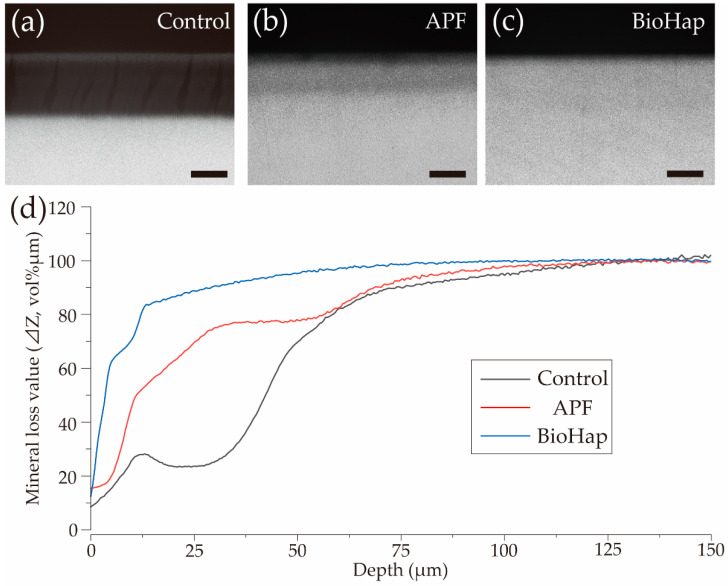
Contact microradiography (CMR) images of enamel cross-sections after acid challenge. CMR cross-sectional images of the reference and experimental surfaces after acid challenge in the (**a**) control, (**b**) APF, and (**c**) BioHap groups. Scale bar is 50 μm. (**d**) Graphical representation of mineral loss value (ΔZ) by tooth depth. Black, red, and blue lines represent the control, APF, and BioHap groups, respectively.

**Figure 7 materials-15-07171-f007:**
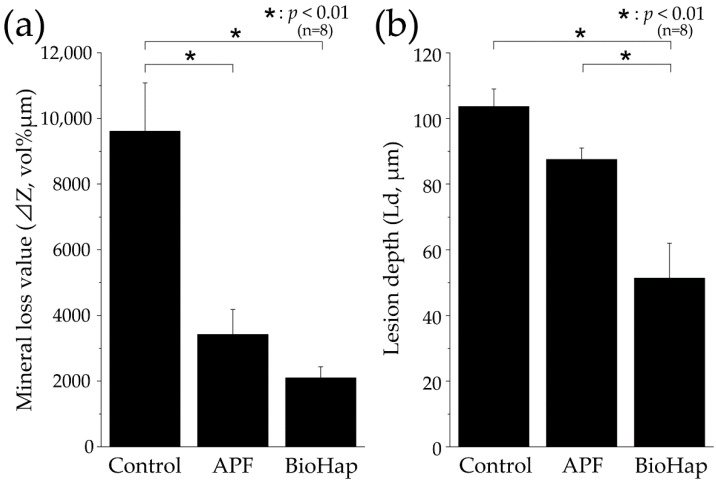
Graphical representations of mineral loss value (ΔZ) and lesion depth (Ld) after acid challenge. (**a**) Graphical representation of mineral loss value (ΔZ, n = 8, * *p* < 0.01). All eight samples were measured, and the mean ± standard deviation values were determined. (**b**) Graphical representation of Ld (n = 8, * *p* < 0.01). The depth of demineralization was determined from the surface before the demineralization experiment to a site with 95% healthy enamel. All eight samples were measured, and the mean ± standard deviation values were determined.

**Table 1 materials-15-07171-t001:** Materials used in this study.

Materials	Composition
BioHap (BIOAPATITE Inc., Shiga, Japan)	(Ca:Mg)_10_(PO_4_)_6_(OH)_2_; powder
Citric acid gel (AIWA Co., Ltd., Osaka, Japan)	5% citric acid, 0.01% malic acid, 5% glycerin, 3% tamarind gum, others (pH 3.0); gel
Acidulated phosphate fluoride (APF)	2% sodium fluoride, 1% phosphoric acid, 9048 ppmF (pH 3.6); solution

## Data Availability

All data are included in the manuscript.

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
