# Peer review of "Improved Enamel Acid Resistance Using Biocompatible Nano-Hydroxyapatite Coating Method"

_materials, 2022, doi:10.3390/ma15207171_

Round 1
Reviewer 1 Report
I have carefully evaluated the manuscript. This manuscript presents a method to improve enamel acid resistance using nano HAp coating. The scientific content and presentation of results are good to be published in Journal. The novelty of the article was stated more precisely.
Some comments are
Usually, eggshell-derived HAp has Mg as an impurity. What is the role of Mg in improving enamel acid resistance? Discuss it.
There is no characterization of eggshell-derived HAp. XRD, FTIR, EDX etc.
Fig. 1: Nanoparticles are not evident from SEM. Take TEM analysis for clarity.
Keyword: “Biomimetic HAp” What is biomimetic HAp? How it is relevant to the present manuscript.
The author stated “but it (eggshell HAp) has not yet been used in dentistry” in the introduction. Some studies are there (DOI: https://doi.org/10.1155/2019/5949232, https://doi.org/10.1111/iej.13644). Comment on it.
Author Response
> We strongly appreciate the reviewer's comment. We are thankful for the time and energy you expended. We will work hard to make the paper better with this revision.
Some comments are
Usually, eggshell-derived HAp has Mg as an impurity. What is the role of Mg in improving enamel acid resistance? Discuss it.
>Thank you very much for providing important comments.
Mg2+ in BioHap exist in the apatite structure replacing Ca2+. It is different from the impurity in HAp formation. Therefore, the coating formed with BioHap and fluoride becomes chemically unstable due to partial replacement of Ca2+ in apatite with Mg2+, and demineralizes preferentially over teeth during acid stimulation. We believe that the coating improves acid resistance because it quickly increases the saturation of calcium ions and phosphate ions during demineralization.
We have add sentence as described below.
Page 13, Line 397-400
Mg2+ in BioHap exist in the apatite structure replacing Ca2+. Therefore, the coating formed with BioHap and fluoride becomes chemically unstable due to partial replacement of Ca2+ in apatite with Mg2+, and demineralizes preferentially over teeth during acid stimulation.
There is no characterization of eggshell-derived HAp. XRD, FTIR, EDX etc.
> In this experiment, we purchased and used high-purity BioHap powder produced by BIOAPATITE Inc. (Shiga, Japan). We omitted compositional analysis of the powders, as the composition is provided by the manufacturer.
Page 2, Line 84-85
BioHap (25 g, BIOAPATITE Inc., Shiga, Japan) and citric gel were mixed in a 1: 2 weight ratio and applied to the surface (Table 1).
Fig. 1: Nanoparticles are not evident from SEM. Take TEM analysis for clarity.
> We agree that additional information on TEM analysis as the reviewer suggested would be valuable. Regrettably, however, because of we don't have the equipment and the experimental technology, we are unable to do the experimentation.
Keyword: “Biomimetic HAp” What is biomimetic HAp? How it is relevant to the present manuscript.
> We appreciate the reviewer's comment on this point. We wanted to use the keyword to express that bioapatite has higher biocompatibility than conventional mineral-derived hydroxyapatite. As the reviewer commented, I removed this keyword because it might not make sense to the reader.
The author stated “but it (eggshell HAp) has not yet been used in dentistry” in the introduction. Some studies are there (DOI: https://doi.org/10.1155/2019/5949232, https://doi.org/10.1111/iej.13644). Comment on it.
> Thank you for pointing me to the reference paper on Eggshell derived nano-hydroxyapatite. We have added two papers to our references (as Ref.22,23). In accordance with the reviewer's comment, we have changed this to following sentence:
Page 2, Line 64-66
Moreover, it is reported to be approximately four times more biocompatible than min-eral-derived apatite, it is still not widely used in dental practice [22,23].
References
- Kaviya B.; Balasubramanian S.K.; Ishwarya G.; Sekar M.; Gurusamy R.; Vijayakumar D.; Anil K. Eggshell derived nano-hydroxyapatite incorporated carboxymethyl chitosan scaffold for dentine regeneration: A laboratory investigation. Int Endod J. 2022, 55, 89-102; doi: 10.1111/iej.13644.
- Sandra J.G.; Luis F.F.; Luis G.S.; Kelly J.D.; Linet Y.C.; José A.L.; Juan C.S.; Adriana P.A. Elaboration and Biocompatibility of an Eggshell-Derived Hydroxyapatite Material Modified with Si/PLGA for Bone Regeneration in Dentistry. Int J Dent. 2019, 2019, 5949232; doi: 10.1155/2019/5949232.
Reviewer 2 Report
This work studies a dental caries prevention method using a bioapatite combining with high-concentration fluoride having acid resistance of enamel. The research content is substantial. However, there are some unsolved experimental phenomena and details to be addressed in this manuscript. Some are as follow:
1.Without affecting the meaning of the picture, the scale bar should be placed on all figures in this manuscript.
2.The language of the whole manuscript needs to be polished largely.
3.The authors should discuss the relevant mechanism in detail.
4.What is the prospect about this biocompatible nano-hydroxyapatite coating?

Author Response
>Thank you very much for providing important comments. We are thankful for the time and energy you expended. Our responses to the referees’ comments are as follow:
1.Without affecting the meaning of the picture, the scale bar should be placed on all figures in this manuscript.
> We appreciate the reviewer's comment on this point. In accordance with the reviewer's comment, we have added scale bar to figure (bottom right of photo). In figure 4, we also changed the color of the scale bar for better visibility (black to white).
2.The language of the whole manuscript needs to be polished largely.
> The paper has been edited and rewritten by an experienced scientific editor, who has improved the grammar and stylistic expression of the paper.
3.The authors should discuss the relevant mechanism in detail.
> We agree with you and have incorporated this suggestion throughout our paper. In accordance with the reviewer's comment, we have added sentence as described below.
Page 13, Line 397-400
Mg2+ in BioHap exist in the apatite structure replacing Ca2+. Therefore, the coating formed with BioHap and fluoride becomes chemically unstable due to partial replacement of Ca2+ in apatite with Mg2+, and demineralizes preferentially over teeth during acid stimulation.
4.What is the prospect about this biocompatible nano-hydroxyapatite coating?
> You have raised an important question. We want to use this method for preventive dentistry. In this experiment, we used enamel as a sample, but we are also considering applying it to dentin. In Japan, the aging population is remarkable, and root caries is rapidly increasing. Moreover, conventional preventive measures are less effective for root caries. Therefore, we think that this method can be used for the prevention of root caries. We will also write a paper on dentin in the near future.
Reviewer 3 Report
The authors are requested to:
1- Add table for the exact chemical composition of the materials tested in the current experiment.
2- In the micro hardness test; Were the authors measuring the micro hardness of the 5 micrometer thickness layer formed by the application of the tested material or was the remineralized enamel micro hardness measured?
3- The authors should declare clearly if their technique mandates the application of fluoride with bioapatite (BioHap) or not.
4- are the authors suggesting the application of bioapatite (BioHap) as an enamel sealer? if so, please add adequate updated references discussing similar agents and similar techniques e.g. 45S5 Bioglass paste is capable of protecting the enamel surrounding orthodontic brackets against erosive challenge.
Effect of citric acid erosion on enamel and dentin and possible protection by a novel bioactive borate adhesive system.
5- Provide fully labelled diagram describing the mechanism of action of their material and compare it to previously suggested mechanism in which the bioactive materials were used in combination with acidic solutions.
6- If possible add another group in which Bio HAP is used without application of fluoride.
Author Response
Thank you for all of your detailed comments and suggestions. We found them quite useful as we approached our revision.
The authors are requested to:
- Add table for the exact chemical composition of the materials tested in the current experiment.
> In accordance with the reviewer's comment, we have added a new table.
Page2, Line 82-85
After ionizing some crystals on the surface by applying citric acid gel (5% citric acid, 0.01% malic acid, 5% glycerin, 3% tamarind gum, pH 3.0, AIWA Co. Ltd. Osaka, Japan), BioHap (25 g, BIOAPATITE Inc., Shiga, Japan) and citric gel were mixed in a 1: 2 weight ratio and applied to the surface (Table 1).
Page 3, Line 97-
Table 1. Materials used in this study.
- In the micro hardness test; Were the authors measuring the micro hardness of the 5 micrometer thickness layer formed by the application of the tested material or was the remineralized enamel micro hardness measured?
> We appreciate the reviewer's comment on this point. We measured acid-damaged surfaces after fluoride application and remineralization. The control group received acid damage without fluoride application. Vickers hardness was not measured for samples that were only coated with BioHap on healthy teeth.
- The authors should declare clearly if their technique mandates the application of fluoride with bioapatite (BioHap) or not.
> The reviewer's comment is correct. You have raised an important question. Although not included in this paper, we performed preliminary experiments with bioapatite-only groups without fluoride. Bioapatite alone could not coat the tooth surface. It is our hypothesis that the formation of the coating requires the formation of calcium fluoride by fluoride after increasing the saturation of calcium and phosphate ions. Therefore, the bioapatite-only group was omitted in this paper.
- are the authors suggesting the application of bioapatite (BioHap) as an enamel sealer? if so, please add adequate updated references discussing similar agents and similar techniques e.g. 45S5 Bioglass paste is capable of protecting the enamel surrounding orthodontic brackets against erosive challenge. Effect of citric acid erosion on enamel and dentin and possible protection by a novel bioactive borate adhesive system.
> Thank you for pointing me to the reference paper on enamel sealer. We have added a paper to our references (as Ref.). In accordance with the reviewer's comment, we have added this to following sentence and reference:
Page 13, Line 427-429
This method physically seals the teeth, and can be used not only for dental caries but also for erosion like bioglass paste and resin [45].
References
- Mona A.A.; Ahmed I.M.; Fahd F.A.; Ahmed S.B. Effect of citric acid erosion on enamel and dentin and possible protection by a novel bioactive borate adhesive system.J Dent. 2022, 124, 104208. doi: 10.1016/j.jdent.2022.104208.
5- Provide fully labelled diagram describing the mechanism of action of their material and compare it to previously suggested mechanism in which the bioactive materials were used in combination with acidic solutions.
> Page 12-13, Line 386-392 describes a comparison between our hypothetical mechanism and previous papers that performed similar procedures using α-TCP. In accordance with the reviewer's comment, we think the diagram is useful. However, we are still hypothesizing the mechanism, and we are not at the stage of creating a diagram due to lack of component analysis data. The mechanism has not been clarified in the reference literature that conducted similar experiments. We plan to clarify the mechanism by conducting component analysis such as XPS and XRD using hydroxyapatite powder.
6-If possible add another group in which Bio HAP is used without application of fluoride.
> We agree that additional information on another group (BioHap without fluoride) as the reviewer suggested would be valuable. As written in question 3, we performed a preliminary experiment on that group. Therefore, the bioapatite-only group was omitted in this paper.
Again, thank you for giving us the opportunity to strengthen our manuscript with your valuable comments and queries. We have worked hard to incorporate your feedback and hope that these revisions persuade you to accept our submission.
Round 2
Reviewer 2 Report
The revised version can be acceptable.
Author Response
The revised version can be acceptable.
> We wish to thank the reviewer for this comment. We are thankful for the time and energy you expended.
Reviewer 3 Report
The manuscript is fine
Author Response
The manuscript is fine.
> We wish to thank the reviewer for this comment. We are thankful for the time and energy you expended.